# Early Maladaptive Schemas and Cognitive Distortions in Adults with Morbid Obesity: Relationships with Mental Health Status

**DOI:** 10.3390/bs7010010

**Published:** 2017-02-28

**Authors:** Felipe Q. da Luz, Amanda Sainsbury, Phillipa Hay, Jessica A. Roekenes, Jessica Swinbourne, Dhiordan C. da Silva, Margareth da S. Oliveira

**Affiliations:** 1The Boden Institute of Obesity, Nutrition, Exercise & Eating Disorders, Sydney Medical School, Charles Perkins Centre, The University of Sydney, NSW 2006, Australia; amanda.salis@sydney.edu.au (A.S.); jess.roekenes@gmail.com (J.A.R.); jessica.swinbourne@sydney.edu.au (J.S.); 2School of Psychology, Faculty of Science, The University of Sydney, NSW 2006, Australia; 3Faculty of Psychology, Pontifical Catholic University of Rio Grande do Sul, Av. Ipiranga 6681, Porto Alegre/RS, CEP 90619-900, Brazil; dhiordanc@gmail.com (D.C.d.S.); marga@pucrs.br (M.d.S.O.); 4Centre for Health Research and School of Medicine, The University of Western Sydney, Locked Bag 1797, Penrith NSW 2751, Australia; p.hay@westernsydney.edu.au

**Keywords:** obesity, morbid obesity, psychology, dysfunctional cognition, mental health

## Abstract

Dysfunctional cognitions may be associated with unhealthy eating behaviors seen in individuals with obesity. However, dysfunctional cognitions commonly occur in individuals with poor mental health independently of weight. We examined whether individuals with morbid obesity differed with regard to dysfunctional cognitions when compared to individuals of normal weight, when mental health status was controlled for. 111 participants—53 with morbid obesity and 58 of normal weight—were assessed with the Mini-Mental State Examination, Young Schema Questionnaire, Cognitive Distortions Questionnaire, Depression, Anxiety and Stress Scale, and a Demographic and Clinical Questionnaire. Participants with morbid obesity showed higher scores in one (insufficient self-control/self-discipline) of 15 early maladaptive schemas and in one (labeling) of 15 cognitive distortions compared to participants of normal weight. The difference between groups for insufficient self-control/self-discipline was not significant when mental health status was controlled for. Participants with morbid obesity showed more severe anxiety than participants of normal weight. Our findings did not show clinically meaningful differences in dysfunctional cognitions between participants with morbid obesity or of normal weight. Dysfunctional cognitions presented by individuals with morbid obesity are likely related to their individual mental health and not to their weight.

## 1. Introduction

Schema Theory proposes that some individuals can develop dysfunctional patterns of beliefs and unhelpful perceptions of the world and themselves [1]. These beliefs and perceptions usually develop during childhood or adolescence as a result of psychologically harmful experiences involving family members or other significant individuals, and for this reason are referred to as early maladaptive schemas. According to this theory, early maladaptive schemas develop in response to unmet core emotional needs, namely: secure attachment to others, autonomy/competence, freedom to express emotions, spontaneity, and realistic limits/self-control. It was previously suggested that as a result of this process, people can develop different types of psychological disorders and engage in a continuum of dysfunctional behaviors [1].

There is some evidence that early maladaptive schemas can relate to dysfunctional eating behaviors. For example, the unhealthy eating behaviors seen in patients with eating disorders were found to be associated with the presence of early maladaptive schemas [2]. One study [3] evaluated the presence of early maladaptive schemas in participants with obesity and found that they showed more severe early maladaptive schemas than participants of normal weight, notably the early maladaptive schemas of social isolation/alienation and defectiveness/shame. Another study [4] found that the early maladaptive schemas of isolation/alienation, emotional inhibition, abandonment/instability and unrelenting standards/hypercriticalness negatively influenced aspects of identity amongst individuals with obesity. Additionally, a higher presence of the following early maladaptive schemas were found amongst adolescents with overweight or obesity in comparison to adolescents of normal weight: social isolation/alienation, defectiveness/shame, emotional deprivation, failure to achieve, dependence/incompetence, and subjugation [5], as well as emotional deprivation, abandonment/instability, subjugation and insufficient self-control/self-discipline [6]. Additionally, another study [7] involving adolescents with overweight found that those who experienced a loss of control over eating had a greater severity of the early maladaptive schemas of social isolation/alienation, abandonment/instability, unrelenting standards/hypercriticalness, mistrust/abuse, failure to achieve and subjugation, in contrast with those that did not experience loss of control over eating.

Another set of dysfunctional cognitive processes, named cognitive distortions, plays an important role in maintaining the negative core beliefs that form early maladaptive schemas, through the perceptual distortion of facts [1]. Cognitive distortions are common thoughts that happen quickly, involuntarily, and in a distorted manner [8]. Some specific types of cognitive distortions have been suggested to be experienced by individuals with obesity [9]. These distorted thoughts occur, for example, when someone thinks that the desire to eat is irresistible (“magnification”), that they are “losers” because they are obese (“labeling”) or that people reject them because they are overweight (“mind reading”). One study found that dichotomous thinking (a type of cognitive distortion) about food, weight and eating was predictive of weight regain, and that a general dichotomous thinking pattern (not necessarily related to food, weight or eating) was an even better predictor of weight regain [10]. Two other studies [11,12] assessed vulnerability to a specific cognitive distortion, namely thought-shape fusion, in participants with obesity and participants of normal weight. This type of cognitive distortion occurs when the imagination of the consumption of high-energy food generates the feeling of being fat and negative moral judgment. In these studies, individuals with obesity were less vulnerable to thought-shape fusion than individuals of normal weight, thus revealing differences in cognitive processes between groups. Studies have further examined the correlation of cognitive distortions with binge eating disorder. A small (*n* = 42) exploratory study [13] reported that participants with obesity, whether or not they had binge eating disorder, showed more cognitive distortions than participants of normal weight. In contrast, another study [14] found that individuals with obesity and comorbid binge eating disorder were more affected by dichotomous thinking than individuals with obesity but without binge eating disorder. The studies above suggest that individuals with obesity, especially those with comorbid binge eating disorder, experience more of some types of cognitive distortions than individuals of normal weight.

Other studies, however, emphasize the relationship between dysfunctional cognitions and mental health status independently of the occurrence of eating or weight disorders. For example, there is evidence that early maladaptive schemas predict depression [15], are associated with complex cases of mood and anxiety disorders [16], and are vulnerability factors for the development of symptoms of depression and anxiety amongst individuals experiencing stressful situations [17]. There are also indications that early maladaptive schemas predict occupational stress [18]. Both early maladaptive schemas and cognitive distortions were found to be significantly associated with emotional problems, namely depression and anxiety [19]. In regards to cognitive distortions specifically, a study found that participants with depression showed strong negative interpretations of metaphors [20]. Cognitive distortions also predict depression and anxiety amongst children and adolescents [21]. Additionally, the cognitive model proposes that cognitive distortions, together with neurobiological correlates, influence how people cope with stressful situations and develop depression [22]. All of the studies discussed in this paragraph show a clear association of the occurrence of dysfunctional cognitions and mental health problems, irrespective of the weight of the participants. Therefore, it is possible that the observation that individuals with obesity experience more early maladaptive schemas or cognitive distortions than individuals of normal weight could be mediated by the fact that participants with obesity frequently experience symptoms of mental illness [23], and not because of their elevated weight.

In summary, dysfunctional thinking styles, known as early maladaptive schemas and cognitive distortions, have been found in individuals with eating disorders, overweight or obesity. However, it is possible that these findings are associated with the mental health status of the participants and not necessarily with their obesity. Our study thus aimed to further examine this issue. Therefore, we sought to clarify whether individuals with morbid obesity show higher levels of dysfunctional cognitions than individuals of normal weight, and if so, if this is related to their individual mental health condition. Ultimately, this understanding could aid in the development of effective psychological and behavioral assessments and subsequent interventions, tailored specifically for individuals with morbid obesity.

## 2. Methods

### 2.1. Ethical Considerations

The study was conducted in accordance with the Declaration of Helsinki. This project was approved by the Research Ethics Committees of the Pontifical Catholic University of Rio Grande do Sul (Brazil) (CAAE: 07888612.4.0000.5336) and the Conceição Hospital Group where the participants were assessed. Informed consent was obtained from participants.

### 2.2. Participants

Participants were included if they had morbid obesity (BMI ≥ 40 kg/m^2^) or normal weight (BMI: 18.5–24.9 kg/m^2^) [24], were aged between 18 and 65 years, and had at least five years of education. The exclusion criterion was low cognitive performance (score ≤ 23) as assessed by the Mini Mental State Examination (MMSE) [25], as this could compromise comprehension and hence the accuracy in answering questionnaires. Two potential participants were excluded from the study because of this criterion.

There were 111 participants in this study, 53 (47.7%) with morbid obesity and 58 (52.3%) of normal weight. Participants with morbid obesity were recruited from the hospital bariatric surgery clinic and were classified as such by the medical team. Participants of normal weight were recruited through advertisements within the hospital. Participants’ weight and height were recorded based on self-report. The groups were comparable with regards to sex, age, education, marital status and economic criteria of participants (see Table 1). Participants were not compensated for their participation in this research.

### 2.3. Questionnaires and Interviews

All questionnaires and interviews were conducted or overseen by the first author, in the bariatric surgery clinic of the hospital. The assessments of participants with morbid obesity occurred before bariatric surgery.

#### 2.3.1. Mini Mental State Examination (MMSE)

The MMSE was used to assess mental state and cognitive deficits in potential participants [25]. The questions in this examination are divided into seven categories to assess specific cognitive functions: time orientation, place orientation, attention, basic calculation, word recognition and memorization, language and visual ability. The scores for examination can range from 0 to 30 points, with scores equal to or higher than 24 indicating normal cognition [25].

#### 2.3.2. Young Schema Questionnaire (YSQ)

The YSQ is a self-report questionnaire that aims to identify the occurrence of early maladaptive schemas. This questionnaire has been used for research into core beliefs associated with psychological disturbances [26]. It is available in both a long and short version (205 and 75 items, respectively). Both versions of the YSQ have good psychometric properties, as indicated by statistically significant internal consistency [1]. Indeed, Cronbach’s alpha is greater than 0.80 for each of the subscales on both versions [27]. The questionnaire’s short form (YSQ-S2) was validated for use in Brazil (Cronbach’s alpha = 0.95) [28] and was used in this study. The YSQ-S2 consists of a 75-item questionnaire assessing 15 types of early maladaptive schemas (groups of 5 items assess each of the 15 schemas). Participants are asked to answer the degree that they emotionally feel that the statements describe them according to the following options: 1—Completely untrue of me; 2—Mostly untrue of me; 3—Slightly more true than untrue; 4—Moderately true of me; 5—Mostly true of me and 6—Describes me perfectly. High scores in the items that relate to a specific early maladaptive schema indicate greater severity.

#### 2.3.3. Cognitive Distortions Questionnaire (CD-Quest)

The CD-Quest is a self-report questionnaire that assesses a combination of the last week’s frequency (no, occasional, much of the time or almost all of the time) and intensity (no, a little, much or very much) with which participants engaged in 15 common types of cognitive distortions [29]. Participants can score from 0 to 5 in each cognitive distortion, with higher scores indicating greater occurrence of the cognitive distortion. This instrument has been validated in Brazil and was found to have robust psychometric properties (Cronbach’s alpha = 0.85) [30].

#### 2.3.4. Depression, Anxiety and Stress Scale (DASS-21)

The DASS-21 measures symptoms of depression, anxiety and psychological stress in clinical and nonclinical groups. It has good internal consistency and concurrent validity [31]. In the current study, the short version with 21 items was used. This instrument has adequate internal consistency, with a Cronbach’s alpha of 0.94 for the depression scale, 0.87 for the anxiety scale and 0.91 for the stress scale [31]. The DASS-21 has been validated for use in Brazil and the following Cronbach’s alpha values were found for the depression, anxiety and stress scales, respectively: 0.92, 0.86 and 0.90 [32]. The DASS-21 provides specific scores for depression, anxiety and stress. Higher scores indicate greater severity of the symptoms.

#### 2.3.5. Demographic and Clinical Questionnaire

Demographic and clinical features were assessed by a structured interview developed for this study. Questions included date of birth, marital status, education level, economic status, and use of psychiatric medications. The Brazilian Economic Criteria were used to classify economic status according to the following categories (highest to lowest affluence): A, B, C, D and E [33], computed from quantification of household assets and income.

### 2.4. Statistical Analysis

Descriptive statistics were used to describe the results in terms of absolute and relative distribution, as well as central tendency and variability measures. Age distribution was compared using the Kolmogorov-Smirnov test [34].

For bivariate analysis of categorical variables, the Pearson’s chi-square test (χ2) was used. Fisher’s exact test was performed in contingency tables where at least 25% of cell values presented an expected frequency of less than 5. Monte Carlo simulation was used when at least one variable had polytomous characteristics. For correlational analysis of continuous variables, the Spearman ranked correlations test (Spearman rho, r_s_) was employed because of non-normality of data. For between-group comparisons of non-parametric ordinal or continuous variables, the Mann-Whitney U test was used.

On data inspection, all early maladaptive schemas and cognitive distortions correlated significantly with levels of depression, anxiety and stress as measured with the DASS-21 (all r_s_ > 0.40, all *p* > 0.001). Thus, Multivariable logistic regression analysis was applied to test the strength of association between the dependent variable (obesity, using binary yes/no, with the reference category being ‘participants with obesity’) and independent variables (early maladaptive schemas and cognitive distortions) that were found to be significant or approaching significance at a level of 5% when univariate tests were performed and adjusted for levels of depression, anxiety and stress. A significance level of *p* < 0.05 was employed for all tests. Analyses were conducted using the SPSS for Windows version 20 (Armonk, NY, USA).

## 3. Results

### 3.1. Early Maladaptive Schemas in Participants with Morbid Obesity Versus Participants of Normal Weight

As shown in Table 2, scores for the early maladaptive schema of insufficient self-control/self-discipline were significantly higher in participants with morbid obesity compared to participants of normal weight, as assessed using the YSQ-S2. High scores on this early maladaptive schema indicate beliefs of insufficient control over emotions or impulses and beliefs of not having enough capacity to deal with boredom or frustration in order to complete tasks [1]. There were no other statistically significant differences between participants with morbid obesity and participants of normal weight with respect to scores on early maladaptive schemas (see Table 2).

### 3.2. Cognitive Distortions in Participants with Morbid Obesity Versus Participants of Normal Weight

Participants with morbid obesity showed a statistical trend (*p* = 0.05) for higher scores on the cognitive distortion of labeling in comparison with participants of normal weight (see Table 3). Labeling is a type of cognitive distortion that occurs when someone gives derogatory or demeaning names to themselves or others (e.g., “I am a failure”) [29]. There were no other statistically significant differences between participants with morbid obesity and participants of normal weight, with respect to scores on cognitive distortions.

### 3.3. Depression, Anxiety and Stress Symptoms in Participants with Morbid Obesity Versus Participants of Normal Weight

No significant differences between groups were found on scores of symptoms of depression or stress. However, participants with morbid obesity showed significantly higher scores on anxiety symptoms in comparison to participants of normal weight (see Table 4). Median and interquartile range were used (instead of mean and standard deviation) to analyze differences between participants with morbid obesity and participants of normal weight, because data on levels of depression, anxiety and stress were highly skewed.

### 3.4. Comparative Analysis by Binary Logistic Regression to Predict Morbid Obesity

Logistic regression analysis revealed that the difference in scores of the early maladaptive schema of insufficient self-control/self-discipline between participants with morbid obesity and participants of normal weight was no longer significant when adjustment was made for levels of depression, anxiety and stress. The difference in scores of the cognitive distortion of labeling between participants with morbid obesity and participants of normal weight however was significant when adjustment was made for levels of depression, anxiety and stress (see Table 5).

### 3.5. Psychiatric Medication Use in Participants with Morbid Obesity Versus Participants of Normal Weight

Data from the demographic and clinical questionnaire showed that the group of participants with morbid obesity had a significantly higher number of users of psychiatric medication at the time of the assessment in comparison to participants of normal weight (28 out of 53 or 54% of participants with morbid obesity versus 8 out of 58 or 14% of the participants of normal weight, χ^2^ = 19.98, *p* < 0.001).

## 4. Discussion

The main aim of this study was to compare the occurrence of early maladaptive schemas and cognitive distortions in participants with morbid obesity versus those of normal weight, and to examine if mental health status could influence potential differences between groups in the occurrence of these dysfunctional cognitions. Higher occurrences of the early maladaptive schema of insufficient self-control/self-discipline and a statistical trend towards higher occurrence of the cognitive distortion of labeling were found in participants with morbid obesity compared to participants of normal weight. However, after controlling for symptoms of depression, of anxiety, and stress, participants with morbid obesity and participants of normal weight did not differ statistically in regards to scores on early maladaptive schemas. These findings support a previous study that found that an individual’s responses in the YSQ-S are influenced by their emotional state while completing the questionnaire [35]. Furthermore, even before controlling for mental health status, the statistical differences of early maladaptive schemas and cognitive distortions found amongst participants with morbid obesity compared to participants of normal weight were small (1 out of 15 types of early maladaptive schemas and 1 out of 15 types of cognitive distortions). These are slight differences that may have been found due to chance. These findings do not indicate significant clinical differences in the occurrence of dysfunctional cognitions between those with morbid obesity and those of a normal weight.

We did not find statistically significant differences between participants with morbid obesity and participants of normal weight in regards to symptoms of depression and stress. However, significant differences were found in the presence of anxiety symptoms between groups. These findings contradict a systematic review and meta-analysis that found an association between obesity and depression, particularly amongst women [36]. The levels of depression symptoms in the participants with morbid obesity in our sample were possibly low since more than half (54%) of our participants with morbid obesity were being treated with psychiatric medication at the time of the assessment. Although stress-induced eating habits seem to have an important role in the development of obesity [37], in our study, no differences in the level of stress symptoms were found between groups. Our findings regarding the higher anxiety symptoms amongst participants with morbid obesity are consistent with the outcomes from a systematic review and meta-analysis that found a positive association between anxiety disorders and obesity [38]. A previous study found that individuals with obesity, especially those with comorbid binge eating disorders, tend to eat in response to unpleasant emotional states [39].

An additional finding of our study is that the participants with morbid obesity used significantly more psychiatric medication than the participants of normal weight. This may have been influenced by the fact that the participants with morbid obesity were patients of the bariatric surgery clinic and therefore were regularly seen by the medical team, and such medical attention did not necessarily occur for participants of normal weight. This finding is compatible with a previous study [40] that found high psychiatric medication use amongst individuals with morbid obesity (40.7% of their sample). A controversial issue regarding the prescription of psychiatric medication for individuals with morbid obesity is the effect of these medicines on weight. A recent systematic review reported that body fat accumulation is a common side effect of psychotropic medication [41]. Therefore, it is possible that the higher use of psychiatric medication contributed to the excess weight of the participants with morbid obesity, albeit this was not the focus of the current study.

The current findings have relevance to clinical practice. Lifestyle interventions aimed at promoting healthy eating habits and appropriate levels of physical activity are routinely recommended for people with morbid obesity, due to their role in reducing the medical complications related to morbid obesity and in improving psychological health [42]. However, further psychological therapy may be required for some individuals with morbid obesity. Their mental health status should be assessed, and those with depression, anxiety and/or psychological distress may be considered for further assessment of early maladaptive schemas and cognitive distortions and these (if present) may need to be addressed with specific psychotherapy, such as cognitive and/or schema therapy [1,8].

Limitations of this study include the use of self-report assessment of early maladaptive schemas and cognitive distortions, as some people may have difficulty identifying their own dysfunctional thoughts [43]. A second limitation is that participants with morbid obesity may have tried to express socially desirable responses in an attempt to allay social stigma [44], or for fear of a psychological evaluation that could deny or delay their referral for bariatric surgery [45] (although they were told that their responses would be confidential). A third limitation of this study is that types of psychiatric medication used by the participants, psychotherapeutic treatment and psychiatric diagnosis were not assessed. Future research in this field should include participants with obesity that are not seeking treatment, and examine causal effects of anxiety symptoms and use of psychiatric drugs amongst individuals with morbid obesity.

## 5. Conclusions

In conclusion, higher occurrence of dysfunctional cognitions (the early maladaptive schema of insufficient self-control/self-discipline and the cognitive distortion of labeling) amongst participants with morbid obesity in comparison to participants of normal weight was small and the early maladaptive schema of insufficient self-control/self-discipline was no longer statistically significant once symptoms of depression, anxiety and stress were controlled for. Dysfunctional cognitions presented by individuals with morbid obesity are probably related to their individual mental health status and not to their weight disorder.

## Figures and Tables

**Table 1 behavsci-07-00010-t001:** Demographic details and body mass index of the participants with morbid obesity versus participants of normal weight.

Variables	Group				*p*
	Morbid Obesity (*n* = 53)		Normal Weight (*n* = 58)		
	*n*	%	*n*	%	
Sex					
Female	41	77.4	45	77.6	>0.999 ^ɸ^
Male	12	22.6	13	22.4	
Age (years)					
Mean ± standard deviation (range)	42.3 ± 9.6 (25–59)		38.7 ± 13.9 (18–65)		0.072 ^£^
Median (interquartile range)	42.0 (35.0–42.5)		38.5 (26.0–52.0)		
Highest education completed					
Primary	24	45.3	24	41.4	0.678 ^¶^
Secondary/Tertiary	29	54.7	34	58.6	
Marital status					
Single	12	22.6	18	31.0	
Married	36	67.9	36	62.1	0.580 ^¶^
Separated/Divorced/Widowed	5	9.4	4	6.9	
Brazilian economic criteria					
Highest affluence	24	45.3	36	62.1	0.089 ^¶^
Lowest affluence)	29	54.7	22	37.9	
Body mass index (kg/m^2^)					
Mean ± standard deviation	48.9 ± 6.3		22.1 ± 1.8		

^ɸ^: Pearson’s chi-square test with continuity correction; ^£^: Students t-test for independent groups; ^¶^: Fisher’s Exact Test for Monte Carlo simulation.

**Table 2 behavsci-07-00010-t002:** Early maladaptive schemas as assessed by the YSQ-S2 in participants with morbid obesity versus participants of normal weight.

YSQ-S	Groups						*p* ^§^
	Morbid Obesity (*n* = 53)			Normal Weight (*n* = 58)			
	Mean	Standard Deviation	Median	Mean	Standard Deviation	Median	
Emotional deprivation	2.5	1.5	2.0	2.0	1.3	1.5	0.08
Abandonment/instability	2.6	1.6	2.0	2.5	1.6	1.6	0.44
Mistrust/abuse	2.4	1.5	2.0	2.2	1.3	1.8	0.58
Social isolation/alienation	2.0	1.4	1.4	1.9	1.3	1.3	0.57
Defectiveness/shame	1.5	1.2	1.0	1.3	0.8	1.0	0.77
Failure	1.7	1.2	1.2	1.6	1.0	1.2	0.72
Dependence/incompetence	1.7	1.1	1.2	1.5	0.7	1.4	0.91
Vulnerability to harm	2.0	1.2	1.6	2.1	1.3	1.6	0.86
Enmeshment	1.7	1.0	1.2	2.0	1.2	1.7	0.16
Subjugation	1.8	1.1	1.4	1.8	1.2	1.2	0.53
Self-sacrifice	4.1	1.5	4.2	3.7	1.6	3.9	0.29
Emotional inhibition	2.5	1.6	1.8	1.9	1.2	1.6	0.11
Unrelenting standards	3.1	1.5	3.0	2.7	1.1	2.5	0.15
Entitlement/grandiosity	2.6	1.4	2.4	2.3	1.2	1.8	0.22
Insufficient self-control/self-discipline	2.5	1.4	2.4	2.0	1.2	1.6	0.01

YSQ-S: Young Schema Questionnaire—short form. ^§^: Values compared using the Mann-Whitney U test.

**Table 3 behavsci-07-00010-t003:** Cognitive distortions as assessed by the CD-Quest in participants with morbid obesity versus participants of normal weight.

CD-Quest (¥)	Group						*p* ^§^
	Morbid Obesity (*n* = 53)			Normal Weight (*n* = 58)			
	Mean	Standard Deviation	Median	Mean	Standard Deviation	Median	
All-or-nothing thinking	1.5	1.9	0.0	1.3	1.7	0.0	0.60
Fortune-telling	1.2	1.7	0.0	1.1	1.5	0.0	0.91
Disqualifying	0.5	1.0	0.0	0.5	1.2	0.0	0.45
Emotional reasoning	1.9	2.0	1.0	1.8	1.8	1.0	0.93
Labeling	1.4	1.7	1.0	0.8	1.3	0.0	0.05
Magnification/minimization	0.8	1.3	0.0	0.7	1.3	0.0	0.67
Mental filter	0.9	1.5	0.0	1.1	1.5	0.0	0.22
Mind reading	1.3	1.5	1.0	1.1	1.6	0.0	0.39
Overgeneralization	1.0	1.5	0.0	1.2	1.6	0.0	0.37
Personalization	0.8	1.4	0.0	0.8	1.2	0.0	0.58
Should statements	2.4	1.9	2.0	1.9	1.7	2.0	0.21
Jumping to conclusions	1.4	1.8	1.0	1.4	1.6	1.0	0.69
Blaming	1.6	1.9	1.0	1.3	1.7	0.0	0.45
What if...?	1.4	1.9	0.0	1.8	1.9	1.0	0.30
Unfair comparisons	1.2	1.8	0.0	1.0	1.5	0.0	0.98
CD-Quest Total	19.1	15.7	16.0	17.7	14.4	15.0	0.74

CD-Quest: Cognitive Distortions Questionnaire. ^§^: Values compared using the Mann-Whitney U test; ¥: Asymmetrically distributed variable.

**Table 4 behavsci-07-00010-t004:** Levels of depression, anxiety and stress scores on the DASS-21 in participants with morbid obesity compared to participants of normal weight.

DASS-21	Group				Z, *p*
	Morbid Obesity (*n* = 53)		Normal Weight (*n* = 58)		
Depression					
Mean ± standard deviation	9.7 ± 10.2 (0–40.0)		7.3 ± 9.9 (0–42.0)		
Median (interquartile range)	6.0 (2.0–14.0)		4.0 (0–10.5)		−1.58, 0.14
Anxiety					
Mean ± standard deviation (range)	12.6 ± 11.9 (0–42.0)		8.1 ± 10.1 (0–42.0)		
Median (interquartile range)	8.0 (2.0–23.0)		4.0 (0–12.5)		−2.37, 0.018
Stress					
Mean ± standard deviation (range)	16.1 ± 13.4 (0–42.0)		13.1 ± 11.1 (0–42.0)		
Median (interquartile range)	14.0 (4.0–30.0)		9.0 (3.5–20.0)		−0.953, 0.34

Z = Z score from the Mann-Whitney U test, conducted on the non-parametric median and interquartile range statistics.

**Table 5 behavsci-07-00010-t005:** Results of multivariable (logistic regression) analysis with presence or absence of morbid obesity as dependent variable * (Model Nagelkerke R^2^ = 0.15).

Predictor (Independent) Variable	Exp (B)	95% Confidence Interval	*p*
Early maladaptive schema: insufficient self-control/self-discipline	0.70	0.47; 1.05	0.09
Cogntive distortion: Labelling	0.69	0.49; 0.96	0.03
Anxiety	0.92	0.85; 0.99	0.03
Depression	1.05	0.97; 1.13	0.25
Stress	1.06	0.99; 1.13	0.10

* The reference category (specified as ‘participants with obesity’) is such that a value of Exp (B) (also referred to as the odds ratio) of less than 1 implies that the predictor variable is higher in participants with obesity.

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
