# Peer review of "Early Maladaptive Schemas and Cognitive Distortions in Adults with Morbid Obesity: Relationships with Mental Health Status"

_behavsci, 2017, doi:10.3390/bs7010010_

Round 1

Reviewer 1 Report

This is a well-written study examining early maladaptive schemas and cognitive distortions among adults with and without morbid obesity.  Significant differences were found between the groups on the early maladaptive schema of insufficient self-control/self-discipline, but this was no longer significant when mental health status was controlled for.  The authors found differences between the two groups on the cognitive distortion of labeling that remained significant when controlling for mental health status.  Overall few differences were found between the two groups.  The authors conclude that dysfunctional cognitions presented by individuals with morbid obesity are related to their mental health status and not to their weight. 

Methods: The authors clarify in the discussion that the questionnaires were completed prior to bariatric surgery, but this should be stated clearly in the Methods section. 

Discussion: Some discussion of the clinical implications of these findings would be helpful.

Author Response

1.      Methods: The authors clarify in the discussion that the questionnaires were completed prior to bariatric surgery, but this should be stated clearly in the Methods section. 

Our response: We corrected this (please see lines 154-155), thank you.

2.      Discussion: Some discussion of the clinical implications of these findings would be helpful.

Our response: We added a discussion of the clinical implication of the findings, thank you. Please see lines 336-344.

Reviewer 2 Report

In their article, “Early maladaptive schemas and cognitive distortions in adults with morbid obesity: relationships with mental health status,” the authors evaluated the relation between distorted cognitions and weight status (i.e., comparing individuals with morbid obesity to individuals with normal weight). The authors found that individuals with obesity had elevated scores on indices of labeling (cognitive distortion) and insufficient self-control/self-discipline (maladaptive schema) compared to individuals with normal weight, but that the effect of the schema was explained by comorbid psychiatric problems. The authors concluded that these effects do not suggest clinically significant distinctions between individuals with varying weight status for problems associated with cognitive distortions or schemas.

This paper aims to extend literature on dysfunctional thinking styles that might explain individual differences attributed to weight status. The authors are commended for pursuing a paper with conclusions of clinically insignificant findings, as a means to contribute to a literature base in this area. However, several areas of the paper warrant comment.

1.      Overall: The overall concern with the paper is that it lacks a rationale for why the authors elected to conduct the study, and why this evaluation has clinical import. How do these findings—and how does embarking on this investigation—improve our understanding of obesity or weight status? It would help for the authors to expand on this in the Introduction and Discussion. Further, given the overall conclusion that these findings lack clinical significance, it would be helpful for the authors to provide ideas about the implications for intervention or for our clinical understanding of obesity.

2.      Methods/Results: The authors selected a significance threshold of p≤.05; however, typically, p<.05 is reported. This decision is notable for the finding on “labeling” that has a result of p=.05, calling into question whether this is a significant result or trend-level finding.

a.       Further, it seems the logistic regression results controlling for DASS scores suggest lower likelihood of labeling for individuals with obesity versus normal weight, as the OR is less than 1. The results should be clarified. Alternatively, if this interpretation is inaccurate, it would be helpful to explain the methodology more clearly for conducting this regression analysis, so the results could be interpreted accurately.

b.      Finally, the authors indicate in the Discussion (line 297) that the significant findings may be due to chance given the number of tests conducted on schemas and distortions. The authors might consider a correction for significance (e.g., Bonferroni correction) to enhance the rigor for achieving significance.

3.      Methods, MMSE: Please indicate if higher scores equate to impaired mental status or to normal functioning.

4.      Methods: It is recommended the authors indicate if individuals’ height and weight (used to group into weight status groups) were based on self-report or objective measurement.

5.      Methods, Demographics and Clinical Questionnaire: It is curious why the authors only assessed for psychiatric medications and not other psychological treatment. It might be helpful to clarify if this is a standard measure and/or why this was the only form of treatment included in the assessment.

6.      Discussion (line 305-308): The authors suggest medication use could have affected the results. The authors could consider controlling for medication use in the regression analyses.

7.      Tables: The authors include several tables in the paper, some of which may be able to be combined, or be removed and instead summarized in the text (e.g., Tables 2-4).   

a.       Also, it would be helpful if the symbols used to explain tests were presented as superscript and not standard size font, to avoid confusion about the numbers.

8.      Table 1: The author might consider grouping some of the subgroups together by demographic status to reduce the small numbers of participants in various cells (e.g., primary versus secondary+tertiary education).

9.      Table 4: The authors might consider removing the mean and standard deviations from Table 4, since the tests were conducted on the median and IQR.

10.  Discussion (line 313): The authors should add the word “comorbid” to the phrase “individuals with obesity, especially those with comorbid binge eating disorder” if the referenced study refers to individuals with both conditions. Current phrasing suggests that binge eating disorder is a variant of obesity.

11.  Throughout the manuscript: The authors do a great job of using people-first language to describe individuals as, for example, individuals with morbid obesity as opposed to obese individuals. However, there are a few sections of the manuscript where the authors use “obese participants” (e.g., lines 50, 56, 60, 335). It is recommended the authors check for consistency in using people-first language.

Author Response

1.      Overall: The overall concern with the paper is that it lacks a rationale for why the authors elected to conduct the study, and why this evaluation has clinical import. How do these findings—and how does embarking on this investigation—improve our understanding of obesity or weight status? It would help for the authors to expand on this in the Introduction and Discussion. Further, given the overall conclusion that these findings lack clinical significance, it would be helpful for the authors to provide ideas about the implications for intervention or for our clinical understanding of obesity.

Our response: We did not mean to imply there were no clinical implications to the findings. We have made some changes in the fifth paragraph of the Introduction (lines 115-120), added a paragraph on clinical implications in the Discussion (lines 336-344), and edited the last paragraph of the Discussion (lines 361-363) regarding the clinical implications of this study.

2.      Methods/Results: The authors selected a significance threshold of p≤.05; however, typically, p<.05 is reported. This decision is notable for the finding on “labeling” that has a result of p=.05, calling into question whether this is a significant result or trend-level finding.

Our response: We thank the reviewer for this comment. There is a known problem with overreliance on absolute cut-offs for p values, as relevant findings may be understated or overstated. The use of p < 0.05 is a convention and as the reviewer points out, may risk Type I errors where there are multiple tests without specific hypotheses. Similarly findings, can be at risk of a Type II errors, i.e. missing a true finding. However, we did hypothesise that each of the cognitive distortions may be related to weight status. Thus we did not make a Bonferroni or similar adjustment.  However, we accept that there is a chance of Type I errors being present in this instance where the p value equalled 0.5, and we have thus adjusted our reporting of this as a statistical trend. See line 250, and 295.

a.      Further, it seems the logistic regression results controlling for DASS scores suggest lower likelihood of labeling for individuals with obesity versus normal weight, as the OR is less than 1. The results should be clarified. Alternatively, if this interpretation is inaccurate, it would be helpful to explain the methodology more clearly for conducting this regression analysis, so the results could be interpreted accurately.

Our response: We have clarified our text around this point. The reason for lack of clarity in our original text was likely the method of entry in the logistic regression model i.e. the reference group outcome was specified in reverse order. We have rerun the regression in the alternate order to check this, and the odds ratio in this instance are > 1. We have not changed the result as presented, but instead added the following note to Table 5 to clarify: “*The reference category specified such that an Exp (B) or odds ratio less than 1 implies the predictor variable is higher  in  participants with obesity.” In the Methods section we have added that the reference category was ‘participants with obesity’ (lines 226-227).

b.      Finally, the authors indicate in the Discussion (line 297) that the significant findings may be due to chance given the number of tests conducted on schemas and distortions. The authors might consider a correction for significance (e.g., Bonferroni correction) to enhance the rigor for achieving significance.

Our response: Please see above response to comment 2.

3.      Methods, MMSE: Please indicate if higher scores equate to impaired mental status or to normal functioning.

Our response: Thank you, this information was added to the Methods (please see line 163).

4.      Methods: It is recommended the authors indicate if individuals’ height and weight (used to group into weight status groups) were based on self-report or objective measurement.

Our response: Thank you, this information was added to the Methods (please see lines 144-145).

5.      Methods, Demographics and Clinical Questionnaire: It is curious why the authors only assessed for psychiatric medications and not other psychological treatment. It might be helpful to clarify if this is a standard measure and/or why this was the only form of treatment included in the assessment.

Our response: Information in this regard was added to a part of the Discussion addressing limitations of the study. Only psychiatric medications were assessed because unfortunately there was no data available on psychological treatments on the Demographics and Clinical Questionnaire (please see line 352).

6.      Discussion (line 305-308): The authors suggest medication use could have affected the results. The authors could consider controlling for medication use in the regression analyses.

Our response: As medication use is correlated with depression, anxiety and stress, we did not include it in the model because including highly correlated independent variables in regression analyses can distort the results.

7.      Tables: The authors include several tables in the paper, some of which may be able to be combined, or be removed and instead summarized in the text (e.g., Tables 2-4).   

Our response: We respectfully disagree. We think that the tables provide relevant information (e.g. types of schemas and cognitive distortions that were examined in Tables 2 and 3, respectively; and mean scores on the DASS-21 on Table 4).

a.       Also, it would be helpful if the symbols used to explain tests were presented as superscript and not standard size font, to avoid confusion about the numbers.

Our response: We changed the symbols used to explain tests to superscript in Tables 1, 2 and 3.

8.      Table 1: The author might consider grouping some of the subgroups together by demographic status to reduce the small numbers of participants in various cells (e.g., primary versus secondary + tertiary education).

Our response: We thank the reviewer for this suggestion and we have separated by primary versus grouped secondary plus tertiary education, we have grouped separated, divorced or widowed together in a common category, and grouped the 2 highest and the 2 lowest levels of economic status together into 2 separate categories. There were still no statistically significant differences, but we have changed Table 1 to report findings from these new groups.

9.      Table 4: The authors might consider removing the mean and standard deviations from Table 4, since the tests were conducted on the median and IQR.

Our response: For ease of use by readers, we did not remove the mean and SD but we have made the following note on the table to clarify: “Z=Z score from the Mann Whitney U test, conducted on the non-parametric median and IQ range statistics”.

10.  Discussion (line 313): The authors should add the word “comorbid” to the phrase “individuals with obesity, especially those with comorbid binge eating disorder” if the referenced study refers to individuals with both conditions. Current phrasing suggests that binge eating disorder is a variant of obesity.

Our response: Thank you, the word “comorbid” was added to the sentence (please see line 321).

11.  Throughout the manuscript: The authors do a great job of using people-first language to describe individuals as, for example, individuals with morbid obesity as opposed to obese individuals. However, there are a few sections of the manuscript where the authors use “obese participants” (e.g., lines 50, 56, 60, 335). It is recommended the authors check for consistency in using people-first language.

Our response: We appreciate your collaboration in promoting people-first language around obesity, and we changed to people-first language throughout the whole manuscript.

Round 2

Reviewer 2 Report

The authors have revised their paper and addressed the comments from the original review.  The paper has been improved with these changes, particularly in terms of explaining the clinical relevance of this study and the results. Also, the note in Table 5 regarding the analyses and interpretation of the results is a helpful addition.